# Diagnosing Failure Modes of Neural Operators Across Diverse PDE Families

**Lennon J. Shikhman**                                                  *lj@shikhman.net*
*Georgia Institute of Technology*

**Reviewed on OpenReview:** *https://openreview.net/forum?id=0S1LWZHQYn*

## Abstract

Neural PDE solvers are increasingly used as learned surrogates for families of partial differential equations, where the key machine learning challenge is not only interpolation on a fixed benchmark distribution but generalization under structured shifts in coefficients, boundary conditions, discretization, and rollout horizon. Yet evaluation is still often dominated by in-distribution test error, making robustness difficult to assess. We introduce a standardized stress-testing framework for neural PDE solvers under deployment-relevant shift. We instantiate it on three representative architectures—Fourier Neural Operators (FNOs), a DeepONet-style model, and convolutional neural operators (CNOs)—across five qualitatively different PDE families: dispersive, elliptic, multi-scale fluid, financial, and chaotic systems. Across 750 trained models, we measure robustness using baseline-normalized degradation factors together with spectral and rollout diagnostics. The resulting comparisons reveal that strong in-distribution accuracy does not reliably predict robustness, and that failure patterns depend jointly on architecture and PDE family. Our results provide a clearer basis for evaluating robustness claims in neural PDE solvers and suggest that function-space generalization under structured shift should be treated as a first-class evaluation target.

## 1  Introduction

Neural PDE solvers learn operators between function spaces, offering fast surrogates for families of partial differential equations rather than repeated solves of individual discretized instances [9; 10]. Architectures such as FNO [16], DeepONet [19], and CNO [26], along with wavelet-, geometry-, and attention-based variants, have achieved strong benchmark performance [33; 17; 15]. But for machine learning, low matched-distribution test error is not the whole problem. The real question is whether these models generalize under the structured shifts that arise in use: changes in coefficients, boundary or terminal conditions, discretization, and rollout horizon.

That question remains under-studied. Current evaluation is still dominated by in-distribution error, limited perturbation analyses, or single-equation benchmarks [31; 7; 23]. This is a serious limitation because recent theory suggests that different neural-operator architectures should fail in different ways. For FNO-style models, robustness depends on truncation, discretization, spectral structure, and trainability [5; 13; 30; 8]. For DeepONet-style models, optimization, generalization, and model size matter, and linear-reconstruction methods can be fundamentally inefficient on discontinuous operators [14; 21; 12]. Boundary variation can also change the effective operator family rather than merely induce a mild test perturbation [29; 20]. Similar concerns appear elsewhere in scientific ML, including documented PINN failure modes and spectral bias under high-frequency or multiscale structure [11; 32; 4].

We address this gap through a stress-testing framework for neural PDE solvers under structured shift. We evaluate three representative architectures—FNO, a DeepONet-style model, and CNO—across five PDE families spanning dispersive, elliptic, fluid, financial, and chaotic regimes: nonlinear Schrödinger, Poisson, Navier–Stokes, Black–Scholes, and Kuramoto–Sivashinsky. We then apply controlled shifts in parameters,

boundary or terminal conditions, resolution, rollout horizon, and input perturbations. This setup lets us ask a concrete ML question: does strong in-distribution performance predict robust operator learning, and if not, how do failures depend on architecture and equation class?

Our contributions are:

**(1)** We formulate robustness in neural PDE solving as a structured generalization problem and introduce a stress-testing framework for evaluating it.

**(2)** We instantiate the framework across FNO, DeepONet-style, and CNO models on five qualitatively different PDE families under a common protocol.

**(3)** We combine degradation-based metrics with spectral and rollout diagnostics to identify where robustness breaks down.

**(4)** We show that robustness is not captured by baseline accuracy alone and does not transfer cleanly across equations or stress conditions.

Overall, the paper argues that robustness claims for neural PDE solvers should be grounded in explicit evaluation under structured shift, not inferred from in-distribution accuracy alone.

## 2   Related Work

Neural operator methods learn mappings between function spaces rather than solution fields tied to a single discretization. Foundational work includes the general neural operator framework of Kovachki et al. [9], the Fourier Neural Operator (FNO) of Li et al. [16], and DeepONet [19]; recent reviews synthesize the field from approximation-theoretic, algorithmic, and numerical-linear-algebra viewpoints [10]. The architecture space now includes Physics-Informed Neural Operators (PINO), which incorporate PDE residual structure [18]; Convolutional Neural Operators (CNOs), which emphasize continuous–discrete consistency and robustness across resolutions and distributions [26]; and operator-style forecasting systems such as FourCastNet [24]. Other variants encode multiscale U-shaped structure, wavelet-localized representations, geometry-aware operators for irregular domains, and attention-based Fourier/Galerkin or transformer architectures [25; 33; 17; 1; 15]. This breadth suggests that robustness should be understood through architecture-specific approximation bias rather than a single notion of model quality.

Recent theory has begun to clarify these biases. For FNOs, existing work studies capacity, generalization, expressivity, trainability, truncation, and discretization effects, including Rademacher bounds, mean-field analysis, and decompositions of statistical, truncation, and grid-induced error [5; 8; 30; 13]. Related variants target regimes where standard spectral parameterizations are stressed, including highly oscillatory operators and hyperbolic conservation laws [36; 6]. For DeepONet, recent work analyzes optimization, generalization, and model-size requirements, while multiscale variants aim to mitigate failures on high-frequency operators [14; 21; 34].

Our setting is also related to domain generalization and domain adaptation under distribution shift. Domain generalization studies generalization from source domains to unseen target domains without target-domain access, while domain adaptation assumes some target-domain access for adaptation [37]. Although these settings differ from operator learning for PDEs, they provide a useful lens because our stress tests also probe generalization beyond the training distribution. Here, the learned objects are solution operators between function spaces, and the shifts are structured changes in coefficients, boundary or terminal conditions, discretization, and rollout horizon rather than generic dataset shifts.

This issue is increasingly important for the emerging foundation-model perspective in scientific machine learning. The long-term goal is to build broadly reusable scientific surrogates that generalize across regimes, problem instances, geometries, and eventually broader classes of PDEs, rather than only interpolate within narrow benchmark distributions. Current PDE surrogate methods remain far from that level of breadth, but the foundation-model agenda makes robustness evaluation more urgent: broadly reusable PDE models require direct measurement of failure modes under structured shift. Our work does not study foundation

PDE models directly; it targets a prerequisite for that agenda by stress-testing learned PDE solvers under changes in parameters, boundary or terminal conditions, resolution, rollout horizon, and input perturbations.

This concern is beginning to appear explicitly in scientific machine learning. Setinek et al. [28] study distribution shift and adaptation for neural surrogates through an unsupervised domain adaptation benchmark. Recent work examines out-of-distribution generalization for PDE surrogate models and neural physics solvers under shifts in initial conditions, PDE parameters, and geometry [22; 35]. Related adaptation ideas have also appeared in physics-informed graph learning for cross-regime power-flow prediction under domain shift [3]. These works establish robustness under shift as a central concern in scientific ML, but they differ from our unified comparative framework across multiple PDE families, neural operator architectures, and deployment-relevant stress conditions.

Understanding when scientific machine learning models fail remains open. In the PINN literature, Krishnapriyan et al. [11] showed that failure can arise from optimization and conditioning difficulties rather than lack of expressivity. Spectral bias remains a persistent limitation when high-frequency or multiscale structure must be recovered [32]. In operator learning, discontinuous and interface-dominated problems expose sharper structural limits: methods with linear reconstruction can be fundamentally inefficient for PDEs with discontinuities, motivating nonlinear reconstruction and discontinuity-aware extensions [12]. More recent discontinuity-focused architectures embed interface structure directly into the learned representation [27]. Together, these results support the broader scientific-ML concern that strong in-distribution accuracy need not imply robustness under qualitatively different solution structure [4].

A particularly important under-emphasized issue is conditional generalization under varying boundary or terminal conditions. Recent work argues that, when such conditions vary across samples, neural PDE solvers are often better interpreted as learning boundary-indexed families of operators rather than a single boundary-agnostic solution map [29]. Complementary work proposes explicit mechanisms for conditioning neural operators on complex boundary data, including learned boundary-to-domain extensions, and shows that boundary sensitivity can dominate performance in elliptic settings [20]. Broader benchmarking efforts have expanded empirical evaluation through curated suites and rollout-centered protocols, including PDEBench, APEBench, and The Well [31; 7; 23]. What remains less developed is a common protocol for probing deployment-relevant failure modes across multiple PDE families, architectures, and stress conditions. Our work addresses this gap by systematically comparing robustness under controlled shifts in parameters, boundary or terminal conditions, resolution, rollout horizon, and input perturbations.

## 3 Methodology

### 3.1 Architectures and Evaluation Design

We study robustness in neural PDE solvers through a standardized stress-testing framework. We evaluate three representative architectures under a common protocol: the Fourier Neural Operator (FNO) [16], DeepONet [19], and a convolutional neural operator (CNO) [26]. These models were selected to span three qualitatively different operator-learning mechanisms rather than to exhaust the architecture space. FNO represents global spectral mixing, DeepONet represents branch–trunk operator approximation with separate encoding of inputs and query locations, and CNO represents localized multi-scale convolution. Together they provide a deliberately diverse set of test cases for instantiating the benchmarking framework and for assessing whether robustness patterns are shared across distinct inductive biases or depend strongly on architectural design.

For each PDE family, we generate paired input–output data from an in-distribution regime. Inputs include the relevant forcing terms, coefficients, initial conditions, payoff functions, or parameter channels, depending on the problem, and outputs are either static solution fields or short-horizon targets for time-dependent systems. For PDEs with known scalar parameters, such as viscosity $\nu$ in Navier–Stokes, nonlinearity $\kappa$ in NLS, or volatility $\sigma$ in Black–Scholes, the parameter is included as an input channel.

The goal is robustness profiling under a fixed evaluation protocol, not a final architecture leaderboard or a comprehensive benchmark of all neural PDE solvers. The primary contribution of the paper is the stress-

testing methodology itself rather than the particular model set. We therefore instantiate the framework on three deliberately different architectures under standardized implementations, a shared data-generation pipeline within each PDE family, and broadly comparable model scales, while allowing minor architecture-specific choices needed for stable optimization. Other operator-learning methods could also be studied within the same framework, but the present design is intended to support a controlled comparison in which differences in stress-test behavior can be interpreted primarily in terms of architectural inductive bias rather than changes in data generation or evaluation procedure.

### 3.2 Stress Tests

After training on the in-distribution regime, each model is evaluated under five deployment-relevant shifts:

(A) **Parameter or coefficient shift:** parameters are moved beyond the training range, such as larger $\kappa$ in NLS, lower viscosity $\nu$ in Navier–Stokes, higher volatility $\sigma$ in Black–Scholes, or rougher coefficients in Poisson.

(B) **Boundary or terminal-condition shift:** models are tested on unseen boundary or payoff families.

(C) **Resolution extrapolation:** models trained at one discretization are evaluated at finer or coarser resolutions, with spectral diagnostics used to localize error across frequencies.

(D) **Long-horizon rollout:** one-step predictors are iterated beyond the training horizon to measure error accumulation and dynamical instability.

(E) **Input perturbation sensitivity:** small perturbations are added to state channels to test local stability under corrupted inputs.

Not every stressor applies to every PDE family, but within each PDE the stress grid is fixed and shared across architectures. All evaluations are performed without fine-tuning on the shifted regime.

### 3.3 Metrics and Multi-Seed Aggregation

We use a deterministic multi-seed protocol. For each architecture–PDE pair, we train 50 independent models, yielding 750 trained models in total. Every trained model is evaluated on the same stress suite.

Our primary summary statistic is the *degradation factor*,

$$D^{(i)} = \frac{E_{\text{stress}}^{(i)}}{E_{\text{base}}^{(i)}},$$

where $E_{\text{base}}^{(i)}$ is the baseline relative $L^2$ error of model $i$ on in-distribution test data and $E_{\text{stress}}^{(i)}$ is the worst-case error over the corresponding stress grid. This measures robustness relative to baseline accuracy rather than in absolute terms.

We do not rely on degradation alone. For each architecture and PDE family, we also report absolute baseline $L^2$ error, rollout growth rate, rollout amplification, frequency-binned spectral error summaries under resolution shift, and 95% confidence intervals across seeds. This allows robustness to be assessed through multiple complementary diagnostics rather than a single summary score.

### 3.4 Experimental Setup

Training and evaluation are deterministic for each seed. For Poisson, we train on 512 samples at resolution $n = 128$ with batch size 8 for 3000 steps. For Black–Scholes, we use 512 samples at $n = 256$, batch size 8, and 3000 steps. For nonlinear Schrödinger, we use 256 samples at $n = 256$ and $n_t = 20$, batch size 4, and 4000 steps. For Navier–Stokes, we use 128 samples at $n = 64$ and $n_t = 20$, batch size 2, and 5000 steps. For Kuramoto–Sivashinsky, we use 256 samples at $n = 128$ and $n_t = 20$, batch size 8, and 3000 steps. All models are trained with Adam at learning rate $10^{-3}$.

Model sizes are architecture-appropriate but broadly comparable. FNO uses width 64 and depth 4, with 16 Fourier modes in 1D and 12×12 modes in 2D. CNO uses width 64 and depth 5. The DeepONet-style model uses width 128 and depth 2. All models use coordinate channels. Baseline evaluation uses 64 in-distribution test samples per seed; for time-dependent PDEs, baseline rollout summaries use 5 rollout steps. Raw seed-level outputs are then aggregated into per-PDE summaries, cross-PDE comparison tables, and paper figures using the same analysis pipeline.

# 4 Results

Within this three-architecture comparison, robustness is not predicted by baseline accuracy alone. Although FNO attains the lowest in-distribution error across all five PDE families, that advantage is not stable under structured shift. Under changes in parameters, boundary or terminal conditions, resolution, rollout horizon, and input perturbations, robustness depends jointly on architecture and equation class, and model rankings frequently change.

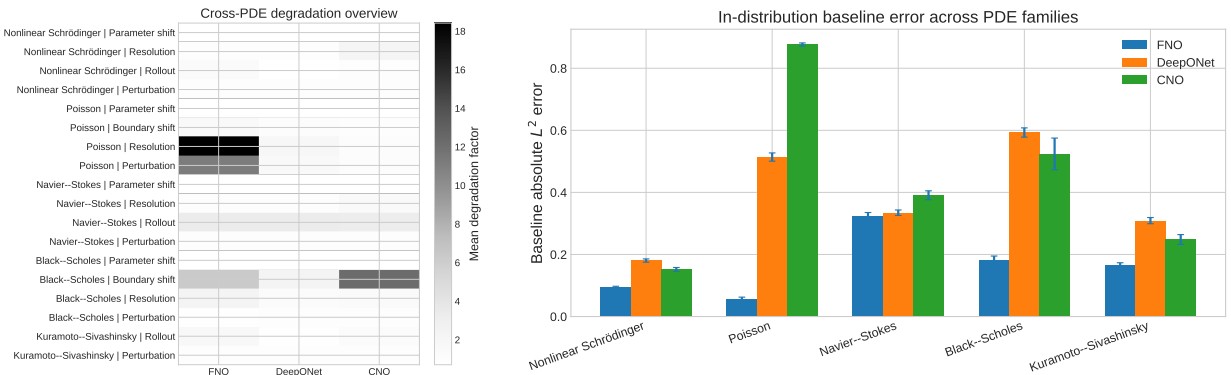

(a) Mean degradation factors across architectures, PDE families, and stress tests. Darker cells indicate more severe degradation.

(b) In-distribution baseline absolute $L^2$ error. FNO is strongest at baseline, but robustness patterns are far less uniform.

Figure 1: Overview of baseline accuracy and robustness across PDE families and architectures.

Figure 1 makes two points clear. First, baseline rankings do not predict robustness under shift. Second, the dominant failure mode is PDE-dependent: Poisson is driven by perturbation and resolution sensitivity, Navier–Stokes by rollout instability, Black–Scholes by payoff-family shift, and nonlinear Schrödinger by parameter extrapolation. These conclusions are supported not only by degradation factors, but also by the accompanying spectral and rollout diagnostics. A useful way to interpret these patterns is through the interaction between PDE structure and architectural inductive bias. FNO emphasizes global spectral mixing and low-frequency operator structure [2], CNO emphasizes localized multiscale convolution together with continuous-discrete consistency [26], and DeepONet emphasizes branch–trunk factorization of operator conditioning and coordinate evaluation [19].

## 4.1 Nonlinear Schrödinger and Poisson

For nonlinear Schrödinger, parameter shift in $\kappa$ is a shared difficulty across all three architectures, but resolution transfer separates them sharply. FNO and DeepONet remain essentially unchanged, while CNO degrades substantially. A PDE-theoretic interpretation is that the dominant stressor here is not locality but how the learned operator responds to changes in the strength of nonlinear dispersive interaction. Since nonlinear Schrödinger remains globally phase-coupled and dispersive, the most important structure is carried by coherent long-range oscillatory modes rather than by sharply localized features. This is broadly consistent with the inductive biases of FNO and DeepONet. FNO is built around global spectral mixing, which is naturally aligned with dispersive wave structure, while DeepONet separates conditioning on the input function from evaluation over the domain, which can remain stable when the operator family is smooth in

resolution. By contrast, CNO's localized multiscale convolutions appear less well matched to this particular resolution transfer setting, even though they are advantageous elsewhere.

Poisson provides the most striking reversal. FNO is clearly best at baseline, yet becomes the least robust model under perturbation and resolution shift. CNO shows the opposite behavior, remaining stable across all stressors despite the worst baseline error. This inversion would be invisible under standard in-distribution evaluation and illustrates how model rankings depend on the intended deployment regime. A PDE-theoretic way to read this reversal is that Poisson is elliptic and globally smoothing, which makes in-distribution interpolation dominated by low-frequency structure. That favors FNO, whose spectral parameterization efficiently captures global smooth components [2]. Under perturbation and resolution shift, however, the challenge is not only recovery of the smooth bulk solution but maintenance of the correct local elliptic response across scales, especially in gradients induced by rougher coefficients or perturbed inputs. There CNO's local multiscale convolutions and its explicit emphasis on continuous-discrete consistency appear better matched to the PDE structure [26]. In other words, the same elliptic smoothing that makes Poisson favorable to spectral interpolation at baseline can mask fragility to local multiscale stress.

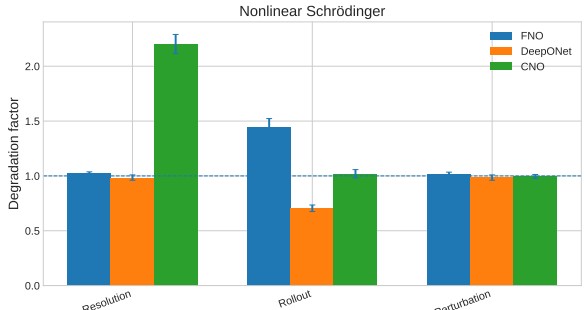

(a) Nonlinear Schrödinger. Parameter shift is broadly difficult, while resolution transfer is strongly architecture-dependent.

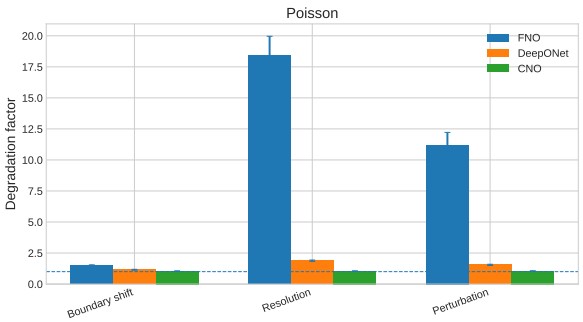

(b) Poisson. FNO is best at baseline but least robust under perturbation and resolution shift, whereas CNO is the most stable.

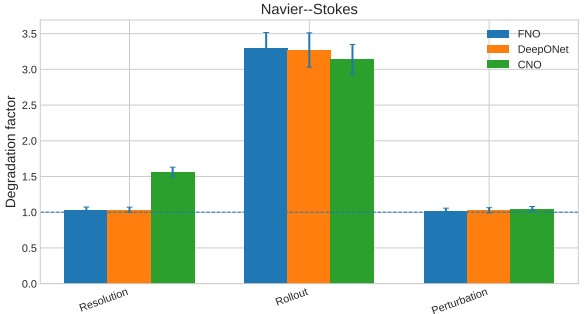

(c) Navier–Stokes. Rollout instability dominates and affects all architectures.

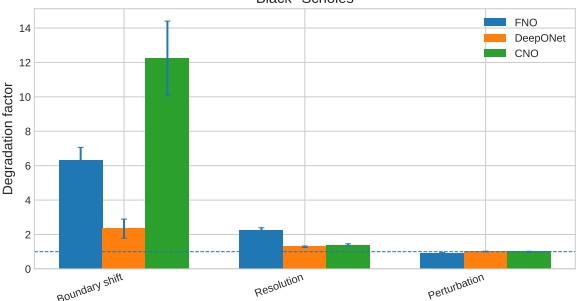

(d) Black–Scholes. DeepONet is most robust under payoff and volatility shift, while CNO exhibits severe degradation.

Figure 2: Degradation-factor summaries across four PDE families. The dominant failure mode differs substantially by equation class, and the strongest baseline model is not always the most robust under stress.

## 4.2 Navier–Stokes and Black–Scholes

Navier–Stokes does not produce a strong ranking inversion. Instead, it reveals a shared limitation: rollout instability. Parameter and perturbation shifts are mild for all models, and resolution transfer only moderately separates them. However, all three degrade substantially under long-horizon rollout. Here the main takeaway is not which model is best, but that current architectures struggle with iterative prediction regardless of

baseline performance. From a PDE perspective, this is a semigroup-composition problem for a nonlinear transport-diffusion system. A one-step predictor only approximates the short-time evolution map, but long-horizon rollout repeatedly composes that approximation. In the vorticity setting, even small local errors are advected forward and re-enter the nonlinear term at the next step, so the relevant issue is not only one-step fit but stability under repeated composition. The fact that FNO, DeepONet, and CNO all deteriorate suggests that this stressor is probing a shared difficulty in learning stable long-time dynamics, despite their different inductive biases. Global spectral coupling, branch–trunk factorization, and local multiscale convolution are all sufficient for short-horizon approximation here, but none by itself guarantees stable autoregressive evolution.

Black–Scholes exhibits a different pattern. The primary challenge is conditional generalization under payoff shift. DeepONet is consistently the most robust model across volatility, payoff, and resolution changes, despite having the worst baseline error. CNO, in contrast, fails severely under payoff shift. This indicates that different PDE families probe different generalization mechanisms: dynamical stability for Navier–Stokes versus functional extrapolation for Black–Scholes. A PDE-theoretic interpretation is that payoff shift changes the terminal-condition family of a backward parabolic problem rather than its underlying dynamics. The Black–Scholes operator maps a payoff at maturity to a smoother solution at earlier times, so robustness under payoff shift depends on how well the architecture represents an operator from an input function family to interior values. This is closely aligned with the DeepONet inductive bias. Its branch net encodes the input function and its trunk net encodes the evaluation location, explicitly matching the structure of a conditional operator $G(u)(y)$ [19]. By contrast, CNO's local multiscale convolutions appear less suited to extrapolating across unseen payoff families, where the main burden is operator conditioning rather than local spatial processing.

### 4.3 Kuramoto–Sivashinsky

Kuramoto–Sivashinsky isolates rollout behavior in a chaotic setting. FNO has the best baseline accuracy but the worst rollout degradation, while DeepONet shows the opposite pattern. This decoupling indicates that one-step accuracy and long-horizon stability are only weakly related, even in a simplified two-stressor setting. From a PDE viewpoint, this is again a composition-of-evolution issue, but now in a chaotic dissipative regime where small phase and amplitude errors rapidly reorganize the solution trajectory. FNO's spectral bias helps at one step because the equation contains strongly structured modal content, but that same advantage does not translate into stable long-horizon rollout once small spectral errors are repeatedly re-injected. DeepONet, despite weaker baseline fit, appears to learn a more rollout-stable conditional map in this setting. As in Navier–Stokes, the key point is that approximation quality at one step and stability under repeated operator composition are distinct properties.

**Takeaways**

Three patterns emerge. First, baseline accuracy is not a reliable proxy for robustness. Second, some weaknesses are shared, most notably instability under repeated composition of learned evolution maps, while others are highly architecture-dependent, such as Poisson resolution sensitivity or Black–Scholes payoff-family shift. Third, robustness varies significantly across PDE families because the stressed operator-theoretic mechanism is problem-dependent. Poisson stresses multiscale elliptic response, Navier–Stokes and Kuramoto–Sivashinsky stress stability of autoregressive evolution, and Black–Scholes stresses generalization across terminal-condition families.

Taken together, these results show that robustness claims do not transfer cleanly across equations or shift types. Evaluating models along a single axis, whether baseline error, one PDE, or one perturbation, can give a misleading picture of performance under realistic deployment conditions.

## 5   Discussion

The main lesson of this study is that robustness in neural PDE solvers is not a single property. It depends jointly on the PDE family, the stress type, and the architecture. If $\mathcal{G}$ denotes the true solution operator

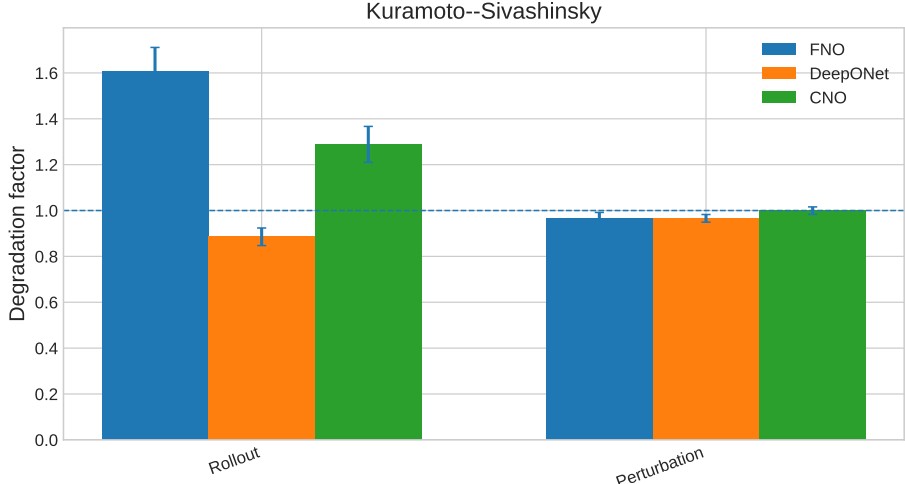

Figure 3: Kuramoto–Sivashinsky. The best baseline model and the most stable rollout model are different.

and $\widehat{\mathcal{G}}$ the learned one, then baseline accuracy mainly measures $\|\widehat{\mathcal{G}} - \mathcal{G}\|$ on one nominal distribution. Our stress tests instead probe how that approximation behaves under structured perturbations of the operator class, the discretization, or repeated composition. In that sense, robustness is closer to a stability question than to an interpolation question. Baseline accuracy therefore gives an incomplete picture: FNO is strongest in-distribution across all five PDE families, yet it is not uniformly the most robust under shift. DeepONet is often more stable under the most consequential shifts, while CNO shows the largest variance across settings.

### 5.1 Two stress-test mechanisms

Two elementary observations help explain why the stress tests reveal behavior that baseline error misses. The first concerns rollout. Let $\Phi$ denote the true one-step evolution map of a time-dependent PDE and let $\widehat{\Phi}$ denote the learned one-step map. Even if $\widehat{\Phi}$ is accurate for one step, long-horizon prediction depends on stability under repeated composition.

**Proposition 1** (Rollout composition bound). *Let $(X, \|\cdot\|)$ be a normed space, let $\Phi : X \to X$ be the true one-step map, and let $\widehat{\Phi} : X \to X$ be a learned one-step predictor. Fix $u_0 \in K \subset X$ and $m \geq 1$, and assume that*

$$u_j = \Phi^j(u_0) \in K, \qquad \widehat{u}_j = \widehat{\Phi}^j(u_0) \in K, \qquad j = 0, \ldots, m-1.$$

*If*

$$\sup_{v \in K} \|\widehat{\Phi}(v) - \Phi(v)\| \leq \varepsilon$$

*and $\widehat{\Phi}$ is $L$-Lipschitz on $K$, then*

$$\|\widehat{\Phi}^m(u_0) - \Phi^m(u_0)\| \leq \varepsilon \sum_{j=0}^{m-1} L^j.$$

The proof is given in Appendix A.1. This proposition formalizes the mechanism behind the Navier–Stokes and Kuramoto–Sivashinsky rollout results. A small one-step error can still produce large long-horizon error if the learned update is not stable under iteration.

The second observation concerns resolution. Let $\mathbb{T}^d$ be the $d$-dimensional torus, let

$$V_N := \operatorname{span}\{e^{ik \cdot x} : \|k\|_\infty \leq N\},$$

and let $P_N$ denote the $L^2(\mathbb{T}^d)$-orthogonal projection onto $V_N$. Let $\widehat{\mathcal{G}}_N$ be a learned finite-resolution approximation of a true solution operator $\mathcal{G}$.

**Proposition 2** (Resolution error decomposition)**.** *Let $(\mathcal{U}, \mathcal{A}, \mu)$ be a probability space of inputs, let $s > 0$, and let*

$$\mathcal{G} : \mathcal{U} \to H^s(\mathbb{T}^d), \qquad \widehat{\mathcal{G}}_N : \mathcal{U} \to V_N$$

*be measurable maps. Define*

$$\mathcal{R}(\widehat{\mathcal{G}}_N) = \mathbb{E}_{u \sim \mu} \|\widehat{\mathcal{G}}_N(u) - \mathcal{G}(u)\|_{L^2}^2$$

*and*

$$\mathcal{R}_N(\widehat{\mathcal{G}}_N) = \mathbb{E}_{u \sim \mu} \|\widehat{\mathcal{G}}_N(u) - P_N \mathcal{G}(u)\|_{L^2}^2.$$

*If*

$$\mathbb{E}_{u \sim \mu} \|\mathcal{G}(u)\|_{H^s}^2 < \infty,$$

*then*

$$\mathcal{R}(\widehat{\mathcal{G}}_N) \leq \mathcal{R}_N(\widehat{\mathcal{G}}_N) + C_{s,d}^2 N^{-2s} \mathbb{E}_{u \sim \mu} \|\mathcal{G}(u)\|_{H^s}^2.$$

*Moreover, let $u_1, \ldots, u_n \overset{\text{iid}}{\sim} \mu$, let $\mathcal{H}_N$ be a class of measurable maps $\mathcal{U} \to V_N$, and define the empirical projected risk by*

$$\widehat{\mathcal{R}}_{N,n}(H) := \frac{1}{n} \sum_{i=1}^{n} \|H(u_i) - P_N \mathcal{G}(u_i)\|_{L^2}^2.$$

*If $\widehat{\mathcal{G}}_N \in \mathcal{H}_N$ is an $\alpha$-approximate empirical risk minimizer, i.e.*

$$\widehat{\mathcal{R}}_{N,n}(\widehat{\mathcal{G}}_N) \leq \inf_{H \in \mathcal{H}_N} \widehat{\mathcal{R}}_{N,n}(H) + \alpha,$$

*and if, with probability at least $1 - \delta$,*

$$\sup_{H \in \mathcal{H}_N} \left| \mathcal{R}_N(H) - \widehat{\mathcal{R}}_{N,n}(H) \right| \leq \Gamma_n(\mathcal{H}_N, \delta),$$

*then with probability at least $1 - \delta$,*

$$\mathcal{R}(\widehat{\mathcal{G}}_N) \leq \inf_{H \in \mathcal{H}_N} \mathcal{R}_N(H) + 2\Gamma_n(\mathcal{H}_N, \delta) + \alpha + C_{s,d}^2 N^{-2s} \mathbb{E}_{u \sim \mu} \|\mathcal{G}(u)\|_{H^s}^2.$$

The proof is given in Appendix A.2. The term $\Gamma_n(\mathcal{H}_N, \delta)$ is left abstract and may be instantiated by standard high-probability uniform convergence bounds under the corresponding boundedness or tail assumptions. This decomposition separates resolution error into a statistical-learning term and a functional-analytic spectral-tail term. It is useful for interpreting the Poisson results: elliptic smoothing can make baseline interpolation largely low-frequency, while perturbation or resolution stress can expose sensitivity to unresolved high-frequency content and local multiscale response. Thus, resolution robustness depends not only on fitted training error but also on the regularity of the PDE solution operator under the stressed regime.

The observed failures can be read through the interaction between PDE structure and architectural inductive bias. FNO imposes a global spectral parameterization, so it is naturally aligned with operators whose dominant behavior is carried by coherent low-frequency modes. CNO imposes local multiscale convolution together with a continuous-discrete consistency bias, so it is better matched to settings where stable local response across scales matters. DeepONet factorizes the approximation of $G(u)(y)$ into dependence on the input function $u$ and dependence on the evaluation coordinate $y$, which is advantageous when conditional operator structure is the main issue.

This helps explain the main reversals. For Poisson, elliptic smoothing makes baseline interpolation largely low-frequency and favorable to FNO, but perturbation and resolution stress probe local multiscale elliptic response, where CNO is more stable. For Black–Scholes, payoff shift changes the terminal-condition family of a backward parabolic problem, so the key issue is conditional generalization across input functions, which is closely aligned with DeepONet. For Navier–Stokes and Kuramoto–Sivashinsky, the main difficulty is repeated composition of a learned short-time map, that is,

$$\widehat{u}_{n+1} = \widehat{\Phi}(\widehat{u}_n),$$

where even small one-step errors can amplify under iteration. The fact that all three models degrade there suggests that long-horizon stability is a harder requirement than one-step accuracy.

Robustness is therefore better understood as a structured profile over operator classes and stress types than as a single scalar attribute. The broader contribution of the paper is to make that profile measurable under a common protocol. The interpretations above should be read accordingly: they are mathematically motivated explanations consistent with the diagnostics, not formal causal proofs.

## 6 Conclusion

We presented a comparative stress-testing evaluation of neural PDE solvers across five PDE families, three architectures, and multiple deployment-relevant shifts. The central result is that robustness is not captured by baseline accuracy alone. Although FNO achieves the lowest in-distribution error across all five PDE families, robustness rankings change substantially under shift. DeepONet is often the most consistently stable model, while CNO shows the greatest variability across equation classes.

More broadly, robustness depends on both the PDE family and the stress condition. Some weaknesses are shared, such as rollout instability in Navier–Stokes, while others are highly architecture-specific, such as Poisson sensitivity to perturbation and resolution or Black–Scholes sensitivity to payoff-family shift. No single benchmark, PDE, or perturbation is therefore sufficient for assessing reliability in neural PDE solvers.

The broader contribution of the paper is the evaluation framework itself. The same stress-based protocol can be extended to other PDE collections, architectures, and application-specific regimes, making it useful beyond the particular benchmark reported here. Taken together, the results suggest that robustness claims for neural PDE solvers should be grounded in explicit evaluation under structured shift rather than inferred from in-distribution accuracy alone.

## Acknowledgements

**Reproducibility.** Code to reproduce all experiments, generate figures, and compute degradation metrics is available at `https://github.com/lennonshikhman/neural-operator-failure-atlas`.

**Computational Resources.** The author gratefully acknowledges Dell Technologies, and in particular the Dell Pro Precision division, for providing computational resources that supported the experiments in this work. All experiments were conducted on a Dell Pro Max T2 workstation equipped with an Intel Core Ultra 9 285K processor, 128 GB of DDR5 ECC memory, and an NVIDIA RTX PRO 6000 Blackwell GPU.

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

# Appendices

## A   Proofs for Auxiliary Stress-Test Mechanisms

This appendix proves the two auxiliary propositions used in Section 5.1. These results are not intended to establish architecture rankings. Instead, they formalize two mechanisms that the stress tests are designed to expose: instability under repeated composition and resolution error induced by unresolved spectral content.

### A.1   Proof of Proposition 1

*Proof.* Let

$$u_j = \Phi^j(u_0), \qquad \widehat{u}_j = \widehat{\Phi}^j(u_0), \qquad j \geq 0,$$

and define the rollout error

$$e_j = \|\widehat{u}_j - u_j\|.$$

By assumption,

$$u_j \in K, \qquad \widehat{u}_j \in K, \qquad j = 0, \ldots, m-1.$$

Since both trajectories start from the same initial condition, $e_0 = 0$. For each $j = 0, \ldots, m-1$,

$$e_{j+1} = \|\widehat{\Phi}(\widehat{u}_j) - \Phi(u_j)\|$$
$$\leq \|\widehat{\Phi}(\widehat{u}_j) - \widehat{\Phi}(u_j)\| + \|\widehat{\Phi}(u_j) - \Phi(u_j)\|.$$

Because $\widehat{\Phi}$ is $L$-Lipschitz on $K$ and $u_j, \widehat{u}_j \in K$,

$$\|\widehat{\Phi}(\widehat{u}_j) - \widehat{\Phi}(u_j)\| \leq L\|\widehat{u}_j - u_j\| = Le_j.$$

The assumed one-step error bound gives

$$\|\widehat{\Phi}(u_j) - \Phi(u_j)\| \leq \varepsilon,$$

since $u_j \in K$. Therefore

$$e_{j+1} \leq Le_j + \varepsilon.$$

Iterating this recursion from $e_0 = 0$ yields

$$e_m \leq \varepsilon(1 + L + \cdots + L^{m-1}) = \varepsilon \sum_{j=0}^{m-1} L^j.$$

Since

$$e_m = \|\widehat{\Phi}^m(u_0) - \Phi^m(u_0)\|,$$

the result follows. $\qquad\square$

**Remark 1.** *The geometric sum may be written explicitly as*

$$\sum_{j=0}^{m-1} L^j = \begin{cases} \dfrac{1 - L^m}{1 - L}, & 0 \leq L < 1, \\ m, & L = 1, \\ \dfrac{L^m - 1}{L - 1}, & L > 1. \end{cases}$$

*Thus, small one-step error does not guarantee stable long-horizon rollout when $L \geq 1$.*

## A.2 Proof of Proposition 2

We first record the standard Fourier spectral-tail estimate.

**Lemma 1** (Fourier spectral tail). *Let*

$$V_N := \operatorname{span}\{e^{ik\cdot x} : \|k\|_\infty \le N\} \subset L^2(\mathbb{T}^d),$$

*and let $P_N : L^2(\mathbb{T}^d) \to V_N$ denote the $L^2(\mathbb{T}^d)$-orthogonal projection onto $V_N$. If $v \in H^s(\mathbb{T}^d)$ for $s > 0$, then there exists a constant $C_{s,d} > 0$ such that*

$$\|(I - P_N)v\|_{L^2} \le C_{s,d} N^{-s}\|v\|_{H^s}.$$

*For the standard Fourier definition of the $H^s$ norm and the cutoff $\|k\|_\infty \le N$, one may take $C_{s,d} = 1$.*

*Proof.* Write

$$v(x) = \sum_{k \in \mathbb{Z}^d} \widehat{v}_k e^{ik\cdot x}.$$

By Parseval's identity,

$$\|(I - P_N)v\|_{L^2}^2 = \sum_{\|k\|_\infty > N} |\widehat{v}_k|^2.$$

If $\|k\|_\infty > N$, then $|k|_2 > N$, hence

$$(1 + |k|_2^2)^s \ge (1 + N^2)^s \ge N^{2s}.$$

Therefore

$$|\widehat{v}_k|^2 \le N^{-2s}(1 + |k|_2^2)^s|\widehat{v}_k|^2.$$

Summing over $\|k\|_\infty > N$ gives

$$\|(I - P_N)v\|_{L^2}^2 \le N^{-2s} \sum_{\|k\|_\infty > N} (1 + |k|_2^2)^s|\widehat{v}_k|^2 \le N^{-2s}\|v\|_{H^s}^2.$$

Taking square roots proves the claim. □

*Proof of Proposition 2.* Let $(\mathcal{U}, \mathcal{A}, \mu)$ be the probability space of inputs, and assume

$$\mathcal{G} : \mathcal{U} \to H^s(\mathbb{T}^d), \qquad \widehat{\mathcal{G}}_N : \mathcal{U} \to V_N$$

are measurable maps, with

$$\mathbb{E}_{u \sim \mu}\|\mathcal{G}(u)\|_{H^s}^2 < \infty.$$

For each $u$, decompose the error as

$$\widehat{\mathcal{G}}_N(u) - \mathcal{G}(u) = \big(\widehat{\mathcal{G}}_N(u) - P_N\mathcal{G}(u)\big) - (I - P_N)\mathcal{G}(u).$$

Since both $\widehat{\mathcal{G}}_N(u)$ and $P_N\mathcal{G}(u)$ lie in $V_N$,

$$\widehat{\mathcal{G}}_N(u) - P_N\mathcal{G}(u) \in V_N.$$

Since $P_N$ is the $L^2$-orthogonal projection onto $V_N$,

$$(I - P_N)\mathcal{G}(u) \in V_N^\perp.$$

Hence the two terms are orthogonal in $L^2(\mathbb{T}^d)$, and Pythagoras gives

$$\|\widehat{\mathcal{G}}_N(u) - \mathcal{G}(u)\|_{L^2}^2 = \|\widehat{\mathcal{G}}_N(u) - P_N\mathcal{G}(u)\|_{L^2}^2 + \|(I - P_N)\mathcal{G}(u)\|_{L^2}^2.$$

Taking expectation over $u \sim \mu$ yields the exact identity

$$\mathcal{R}(\widehat{\mathcal{G}}_N) = \mathcal{R}_N(\widehat{\mathcal{G}}_N) + \mathbb{E}_{u \sim \mu} \|(I - P_N)\mathcal{G}(u)\|_{L^2}^2.$$

Applying Lemma 1 with $v = \mathcal{G}(u)$ gives

$$\|(I - P_N)\mathcal{G}(u)\|_{L^2}^2 \leq C_{s,d}^2 N^{-2s} \|\mathcal{G}(u)\|_{H^s}^2.$$

Taking expectations yields

$$\mathcal{R}(\widehat{\mathcal{G}}_N) \leq \mathcal{R}_N(\widehat{\mathcal{G}}_N) + C_{s,d}^2 N^{-2s} \mathbb{E}_{u \sim \mu} \|\mathcal{G}(u)\|_{H^s}^2.$$

For the learning-theoretic extension, let $\mathcal{H}_N$ be a class of measurable maps $\mathcal{U} \to V_N$, and let $u_1, \ldots, u_n \overset{\text{iid}}{\sim} \mu$. Define the empirical projected risk by

$$\widehat{\mathcal{R}}_{N,n}(H) := \frac{1}{n} \sum_{i=1}^n \|H(u_i) - P_N \mathcal{G}(u_i)\|_{L^2}^2.$$

Assume $\widehat{\mathcal{G}}_N \in \mathcal{H}_N$ is an $\alpha$-approximate empirical risk minimizer, i.e.

$$\widehat{\mathcal{R}}_{N,n}(\widehat{\mathcal{G}}_N) \leq \inf_{H \in \mathcal{H}_N} \widehat{\mathcal{R}}_{N,n}(H) + \alpha.$$

Assume also that, with probability at least $1 - \delta$,

$$\sup_{H \in \mathcal{H}_N} \left| \mathcal{R}_N(H) - \widehat{\mathcal{R}}_{N,n}(H) \right| \leq \Gamma_n(\mathcal{H}_N, \delta),$$

where $\Gamma_n(\mathcal{H}_N, \delta)$ denotes an abstract uniform convergence bound.

Then

$$\mathcal{R}_N(\widehat{\mathcal{G}}_N) \leq \widehat{\mathcal{R}}_{N,n}(\widehat{\mathcal{G}}_N) + \Gamma_n(\mathcal{H}_N, \delta).$$

Using approximate empirical risk minimization,

$$\widehat{\mathcal{R}}_{N,n}(\widehat{\mathcal{G}}_N) \leq \inf_{H \in \mathcal{H}_N} \widehat{\mathcal{R}}_{N,n}(H) + \alpha.$$

Applying the same uniform convergence bound once more,

$$\inf_{H \in \mathcal{H}_N} \widehat{\mathcal{R}}_{N,n}(H) \leq \inf_{H \in \mathcal{H}_N} \mathcal{R}_N(H) + \Gamma_n(\mathcal{H}_N, \delta).$$

Combining the last three displays gives

$$\mathcal{R}_N(\widehat{\mathcal{G}}_N) \leq \inf_{H \in \mathcal{H}_N} \mathcal{R}_N(H) + 2\Gamma_n(\mathcal{H}_N, \delta) + \alpha.$$

Substituting this into the previous bound yields

$$\mathcal{R}(\widehat{\mathcal{G}}_N) \leq \inf_{H \in \mathcal{H}_N} \mathcal{R}_N(H) + 2\Gamma_n(\mathcal{H}_N, \delta) + \alpha + C_{s,d}^2 N^{-2s} \mathbb{E}_{u \sim \mu} \|\mathcal{G}(u)\|_{H^s}^2.$$

This proves the proposition. $\qquad\square$

**Remark 2.** *The term $\Gamma_n(\mathcal{H}_N, \delta)$ is left abstract. In concrete instantiations, it may come from standard high-probability uniform convergence results, such as Rademacher-complexity or covering-number bounds, under the corresponding boundedness or tail assumptions.*

## B  Detailed Degradation Statistics

Table 1: Degradation summary across PDE families and architectures. Each architecture–PDE entry aggregates 50 independent runs; values above 1 indicate increased error under stress.

| Architecture | Stress test | Mean $\pm$ Std | 95% CI |
|---|---|---|---|
| **Poisson Equation** | | | |
| FNO | Parameter shift ($a$ scale) | $1.994 \pm 0.449$ | $[1.870, 2.118]$ |
| | Boundary shift | $1.498 \pm 0.194$ | $[1.444, 1.551]$ |
| | Resolution extrapolation | $18.417 \pm 5.572$ | $[16.872, 19.961]$ |
| | Input perturbation | $11.199 \pm 3.682$ | $[10.178, 12.219]$ |
| DeepONet | Parameter shift ($a$ scale) | $1.265 \pm 0.286$ | $[1.186, 1.344]$ |
| | Boundary shift | $1.155 \pm 0.129$ | $[1.119, 1.190]$ |
| | Resolution extrapolation | $1.882 \pm 0.195$ | $[1.828, 1.936]$ |
| | Input perturbation | $1.545 \pm 0.184$ | $[1.494, 1.596]$ |
| CNO | Parameter shift ($a$ scale) | $1.087 \pm 0.136$ | $[1.049, 1.124]$ |
| | Boundary shift | $1.049 \pm 0.035$ | $[1.039, 1.058]$ |
| | Resolution extrapolation | $1.051 \pm 0.018$ | $[1.046, 1.056]$ |
| | Input perturbation | $1.055 \pm 0.024$ | $[1.048, 1.062]$ |
| **Nonlinear Schrödinger Equation** | | | |
| FNO | Nonlinearity shift ($\kappa$) | $3.872 \pm 0.961$ | $[3.605, 4.138]$ |
| | Resolution extrapolation | $1.022 \pm 0.053$ | $[1.008, 1.037]$ |
| | Long-horizon rollout | $1.449 \pm 0.265$ | $[1.376, 1.523]$ |
| | Input perturbation | $1.020 \pm 0.052$ | $[1.005, 1.034]$ |
| DeepONet | Nonlinearity shift ($\kappa$) | $2.227 \pm 0.503$ | $[2.087, 2.366]$ |
| | Resolution extrapolation | $0.985 \pm 0.088$ | $[0.960, 1.009]$ |
| | Long-horizon rollout | $0.705 \pm 0.108$ | $[0.675, 0.735]$ |
| | Input perturbation | $0.985 \pm 0.088$ | $[0.960, 1.009]$ |
| CNO | Nonlinearity shift ($\kappa$) | $2.618 \pm 0.610$ | $[2.448, 2.787]$ |
| | Resolution extrapolation | $2.203 \pm 0.315$ | $[2.115, 2.290]$ |
| | Long-horizon rollout | $1.020 \pm 0.136$ | $[0.982, 1.057]$ |
| | Input perturbation | $0.995 \pm 0.064$ | $[0.978, 1.013]$ |
| **Navier–Stokes Equation** | | | |
| FNO | Viscosity shift ($\nu$) | $1.028 \pm 0.141$ | $[0.989, 1.067]$ |
| | Resolution extrapolation | $1.034 \pm 0.138$ | $[0.996, 1.072]$ |
| | Long-horizon rollout | $3.296 \pm 0.792$ | $[3.077, 3.516]$ |
| | Input perturbation | $1.018 \pm 0.139$ | $[0.979, 1.057]$ |
| DeepONet | Viscosity shift ($\nu$) | $1.036 \pm 0.131$ | $[1.000, 1.073]$ |
| | Resolution extrapolation | $1.035 \pm 0.130$ | $[0.999, 1.071]$ |
| | Long-horizon rollout | $3.271 \pm 0.866$ | $[3.031, 3.511]$ |
| | Input perturbation | $1.027 \pm 0.131$ | $[0.991, 1.063]$ |
| CNO | Viscosity shift ($\nu$) | $1.056 \pm 0.111$ | $[1.026, 1.087]$ |
| | Resolution extrapolation | $1.562 \pm 0.248$ | $[1.493, 1.630]$ |
| | Long-horizon rollout | $3.138 \pm 0.759$ | $[2.928, 3.348]$ |
| | Input perturbation | $1.050 \pm 0.111$ | $[1.019, 1.081]$ |
| **Black–Scholes Equation** | | | |
| FNO | Volatility shift ($\sigma$) | $2.068 \pm 0.656$ | $[1.886, 2.250]$ |
| | Payoff structure shift | $6.308 \pm 2.733$ | $[5.550, 7.065]$ |
| | Resolution extrapolation | $2.236 \pm 0.552$ | $[2.083, 2.389]$ |
| | Input perturbation | $0.912 \pm 0.170$ | $[0.864, 0.959]$ |
| DeepONet | Volatility shift ($\sigma$) | $1.272 \pm 0.118$ | $[1.239, 1.304]$ |
| | Payoff structure shift | $2.338 \pm 2.005$ | $[1.782, 2.894]$ |
| | Resolution extrapolation | $1.287 \pm 0.154$ | $[1.244, 1.330]$ |
| | Input perturbation | $1.012 \pm 0.059$ | $[0.995, 1.028]$ |
| CNO | Volatility shift ($\sigma$) | $1.357 \pm 0.203$ | $[1.301, 1.413]$ |
| | Payoff structure shift | $12.239 \pm 7.805$ | $[10.076, 14.403]$ |
| | Resolution extrapolation | $1.395 \pm 0.229$ | $[1.331, 1.458]$ |
| | Input perturbation | $0.997 \pm 0.096$ | $[0.970, 1.024]$ |
| **Kuramoto–Sivashinsky Equation** | | | |
| FNO | Long-horizon rollout | $1.607 \pm 0.376$ | $[1.503, 1.711]$ |
| | Input perturbation | $0.967 \pm 0.091$ | $[0.941, 0.992]$ |
| DeepONet | Long-horizon rollout | $0.886 \pm 0.138$ | $[0.847, 0.924]$ |
| | Input perturbation | $0.966 \pm 0.060$ | $[0.949, 0.983]$ |
| CNO | Long-horizon rollout | $1.288 \pm 0.283$ | $[1.210, 1.367]$ |
| | Input perturbation | $1.000 \pm 0.059$ | $[0.983, 1.016]$ |

