# OpenReview forum: "Diagnosing Failure Modes of Neural Operators Across Diverse PDE Families"
_TMLR — Accepted by TMLR_

### Review · Reviewer_a8YP · 2026-03-12

**Summary Of Contributions:**

This paper presents a systematic empirical study of the robustness and failure modes of Fourier Neural Operators (FNOs). The authors design a stress-testing framework to evaluate FNOs under various deployment-relevant scenarios, including parameter shifts, boundary or terminal condition changes, resolution extrapolation, long-horizon rollouts, and input perturbations. Experiments are conducted on five representative PDE families with large-scale multi-seed evaluations. The results highlight common failure patterns of neural operators, such as sensitivity to distribution shifts, difficulty representing high-frequency components, and error accumulation in long-horizon predictions.

**Strengths**
 - The paper provides a systematic evaluation of FNO robustness across multiple PDE families and deployment scenarios. The experimental study is conducted at a relatively large scale (200 models per PDE), which helps ensure that the reported observations are statistically meaningful and not tied to specific random seeds.

**weaknesses**
 - The paper does not provide new insights into the behavior of FNO. The main observations reported in the study—such as FNOs struggling with high-frequency structures, limited generalization under distribution shifts, and error accumulation during multi-step rollouts—are already widely recognized limitations in the neural operator literature (including FNO and other approaches such as DeepONet). As a result, the paper largely confirms existing community understanding through experiments rather than offering new conceptual insights or hypotheses.
 - The paper also does not propose new methods or concrete strategies to address the identified issues. The discussed limitations have already been studied in prior work, where mitigation strategies have been explored. For example, techniques such as injecting perturbations during training or training on short trajectory windows are commonly used to improve long-horizon stability in neural surrogate models. However, the paper does not discuss these directions, nor does it provide suggestions or insights on how the identified failure modes might be mitigated. Consequently, the work remains largely diagnostic and provides limited guidance for improving neural operator models in practical deployment scenarios.

**Audience:**

No

**Audience Explanation:**

For researchers working in AI4PDEs or neural operator learning, the main findings reported in this paper largely reflect limitations that are already widely recognized in the community. Issues such as difficulties in representing high-frequency structures, limited robustness under distribution shifts, and error accumulation in long-horizon rollouts are commonly known characteristics of neural operator models. As a result, while the paper provides a systematic empirical verification of these behaviors, it is unlikely to provide substantially new insights or findings for researchers already familiar with this area.

**Broader Impact Concerns:**

I do not identify specific ethical or societal concerns associated with this work.

**Claims And Evidence:**

Yes

**Claims Explanation:**

Although the main observations discussed in the paper largely reflect limitations that are already widely recognized in the neural operator literature, the paper does provide substantial empirical evidence to support these claims. In particular, the authors conduct large-scale multi-seed experiments across five different PDE families and systematically evaluate multiple stress scenarios, including parameter shifts, boundary or terminal condition changes, resolution extrapolation, long-horizon rollouts, and input perturbations. These experiments provide consistent empirical support for the behaviors described in the paper.

**Requested Changes:**

As discussed above, I find the main limitation of the paper to be the lack of clear contribution. Even for a venue like TMLR, which does not necessarily emphasize novelty, a submission is generally expected to provide either new insights into known phenomena or meaningful directions for addressing existing limitations. In its current form, the paper mainly provides empirical verification of behaviors that are already well understood in the neural operator literature, without offering new conceptual insights or proposing approaches to mitigate these limitations.

To substantially strengthen the work, the authors would need to go beyond empirical confirmation and provide either deeper analysis that leads to new insights about these failure modes, or concrete methodological ideas for addressing them. At present, the paper appears to be relatively far from achieving these goals. For this reason, it is difficult for me to enumerate specific technical revisions that would resolve the concerns, as the issue primarily lies in the overall contribution and positioning of the work.

---

> ### Author Response · Authors · 2026-03-12
> **Response to reviewer a8YP**
>
> Thanks for your thoughtful review. While we agree that the behaviors discussed in the paper are widely recognized in practice when working with FNOs, the goal of the paper isn't to claim these behaviors are new, but to systematically characterize them across various PDE families under a unified experimental framework and to examine quantitatively how they affect model performance across qualitatively different classes of PDEs.
>
> Much of the existing literature evaluates neural operators on individual PDE benchmarks under in-distribution conditions, often with limited analysis of robustness. Prior work such as Krishnapriyan et al. (2021) has studied failure modes in scientific machine learning models, particularly for physics-informed neural networks, but comparable systematic analyses for neural operator architectures remain relatively limited. In particular, there appears to be little work that evaluates robustness across multiple PDE families under a consistent set of stress scenarios. Our work aims to address this gap.
>
> To that end, the paper introduces a suite of stress tests designed to evaluate neural operators under deployment-relevant distribution shifts, including parameter variation, boundary or terminal condition changes, resolution extrapolation with spectral analysis, long-horizon rollouts, and input perturbations. These tests are applied consistently across five PDE systems, which helps reveal which behaviors appear consistently and which depend more strongly on the structure of the underlying equation. We also want to address the reviewer's claim that the paper offers no mitigation discussion. Sections 4 and 5 explicitly discuss several directions for addressing the identified failure modes. While we do not implement these approaches, as the paper's contribution is explicitly diagnostic rather than methodological, we do not agree with this characterization.
>
> To ensure statistical reliability of the results, the experiments use multi-seed evaluation with 200 independently trained models per PDE, totaling 1000 trained models. This large-scale evaluation allows us to quantify the distribution of failure severity across random initializations rather than relying on a small number of runs. We view the main contribution of the paper as a large-scale empirical robustness benchmark for FNO models. While many researchers may informally recognize these limitations, we are not aware of prior work that systematically quantifies them across multiple PDE families within a unified evaluation framework.
>
> We will revise the manuscript to emphasize this framing more clearly and position the work as a robustness evaluation framework for neural operator models. If the reviewer is aware of prior work performing a similar cross-PDE robustness study, we would appreciate a citation so we can clarify the distinction in the revision.

---

> > ### Comment · Reviewer_a8YP · 2026-03-15
> >
> > As noted in my original review, a submission is generally expected to provide either new insights into known phenomena or meaningful directions for addressing existing limitations. If the paper does not propose approaches to address the identified issues, then it should aim to provide clear new insights. In its current form, the paper mainly performs a large number of systematic experiments and reports empirical observations, but it is still unclear what the primary conceptual contribution is.
> >
> > The rebuttal mentions that the work “helps reveal which behaviors appear consistently and which depend more strongly on the structure of the underlying equation”. If the authors view this as the key contribution, it would be important to state this clearly in the abstract and introduction, and to summarize the findings explicitly in the Results or Discussion sections (for example through a clearly structured paragraph, bullet points, or a comparison table). At present, these conclusions mainly appear implicitly within individual experimental discussions, which makes them read more like additional empirical observations rather than the central contribution of the paper.
> >
> > Even with clearer presentation, I am not fully convinced that these observations constitute a strong contribution. As noted earlier, cross-PDE behaviors are already widely recognized in the community. Similarly, claims about PDE-specific behavior would require more direct comparisons to support them. For example, if the paper argues that elliptic PDEs are particularly sensitive to boundary conditions, reviewer would expect comparisons with parabolic or hyperbolic PDEs under similar boundary shifts. Such analyses are currently not included. Nevertheless, improving the clarity of how the authors frame these conclusions would at least help reviewers better evaluate the intended contribution.
> >
> > Finally, regarding mitigation discussion: if proposing solutions is not a primary goal of the paper, that is acceptable. However, in that case the abstract should avoid statements such as “actionable insights for improving robustness in operator learning.” or similar claims, since the current discussion does not provide concrete strategies that would justify this characterization.

---

> > > ### Author Response · Authors · 2026-03-15
> > > **Response 2 to reviewer a8YP**
> > >
> > > Thanks again for your specific and clear feedback. I appreciate you having taken the time to provide such thorough suggestions.
> > >
> > > Once I begin revisions I’ll update the manuscript to clarify the intended contribution of the paper and adjust the phrasing in the abstract and discussion such that it’s more in-line with the scope of the work.

---

> ### Author Response · Authors · 2026-04-11
> **Response to Reviewer a8YP (Revision Update)**
>
> Thank you again for your detailed and constructive feedback. We have made substantial revisions to address your concerns, particularly regarding the clarity of the paper’s contribution.
>
> Most importantly, we expanded the scope beyond a single-model study by incorporating two additional neural operator architectures (DeepONet-style and CNO) within the same stress-testing framework. This allows us to move beyond FNO-specific observations and directly analyze which failure modes are shared across architectures versus those that are architecture-dependent.
>
> In addition, we made significant changes to the framing of the paper to address the concern that the contribution was unclear. The revised manuscript now explicitly positions the work as a comparative diagnostic evaluation framework, rather than a collection of empirical observations. In particular, we clarify that the central contribution is not the individual failure modes themselves, but the structured, cross-PDE and cross-architecture characterization of robustness under a unified set of stress tests.
>
> To make this clearer, we revised the abstract, introduction, and discussion to explicitly state this objective and to present the main conclusions as consolidated takeaways. These now highlight how robustness behavior varies systematically across PDE families and stress conditions, and how different architectures exhibit distinct robustness profiles under the same evaluation protocol.
>
> We also refined the wording throughout to avoid overstating novelty in individual observations and to consistently reflect the diagnostic scope of the work.
>
> We appreciate your feedback, which directly led to a clearer and more focused presentation of the paper’s contribution.

---

### Review · Reviewer_uEgH · 2026-03-19

**Summary Of Contributions:**

This paper presents a systematic stress-testing framework for analyzing the failure modes of Fourier Neural Operators (FNOs) across a diverse set of PDE families. Instead of focusing solely on in-distribution accuracy, the authors design controlled experiments under various distribution shifts, including parameter changes, boundary condition variations, resolution extrapolation, and long-horizon rollouts. The work provides useful empirical insights into common failure patterns such as spectral bias and error accumulation.

A key strength of the paper is the breadth of evaluation across different PDE types and stress scenarios. However, the scope of the analysis is currently limited to FNO, and the connection to broader neural operator literature and alternative approaches(and also for the references) remains insufficiently explored.

**Additional Comments:**

Addressing the above concerns—particularly expanding the scope beyond FNO, improving the discussion of related work, and strengthening the evaluation methodology—would further enhance the contribution and make the findings more broadly applicable.

**Audience:**

Yes

**Audience Explanation:**

Understanding the robustness and failure modes of neural operator models is an important and timely topic for the TMLR community. The proposed stress-testing framework provides a useful perspective that goes beyond standard benchmark evaluations and highlights practical limitations of current approaches. The findings could be particularly valuable for researchers working on improving the generalization, stability, and reliability of neural operators, especially in out-of-distribution settings and long-time prediction tasks.

**Broader Impact Concerns:**

No significant ethical concerns are identified.

**Claims And Evidence:**

No

**Claims Explanation:**

The empirical evidence is generally convincing and well-structured. The stress-testing framework is clearly designed, and the experiments are extensive across multiple PDE families. The observations regarding error amplification under distribution shifts and the concentration of errors in high-frequency modes are well supported by the presented results. However, there are several aspects where the analysis could be further strengthened. In particular, the study focuses exclusively on FNO, which limits the generality of the claims. Including additional neural operator models would help validate whether the identified failure modes are intrinsic to FNO or more broadly shared across operator learning methods.

**Requested Changes:**

## Extending the analysis beyond FNO
While the current analysis provides useful insights, focusing solely on FNO limits the novelty and general applicability of the work. It would significantly strengthen the paper to include additional neural operator models, such as DeepONet and other recent variants, within the same stress-testing framework. This would allow for a clearer understanding of whether the observed failure modes are model-specific or shared across different architectures, and would provide a more comprehensive comparison of strengths and weaknesses.
In addition, when presenting failure cases, it would be helpful to include visualizations comparing predicted solutions and ground truth. Showing where and how the errors occur spatially would greatly improve interpretability and help readers better understand the nature of these failure modes.

## Related work coverage
The current manuscript includes only a limited number of references, which does not fully reflect the recent developments in neural operator research. There has been substantial progress in model design, theoretical analysis, and robustness of neural operators. Expanding the related work section to include more recent contributions would better contextualize the proposed study and clarify its novelty.

## Evaluation metric: degradation factor
The paper primarily relies on the degradation factor as the main evaluation metric. While this metric is useful for measuring robustness, it is not entirely clear whether it is sufficient to capture all aspects of model behavior under different stress conditions. The authors are encouraged to discuss the rationale behind this choice and consider complementing it with additional metrics.
In particular, different stress scenarios (e.g., distribution shifts, resolution changes, long-horizon rollouts) may require different evaluation perspectives. Exploring alternative or complementary metrics could provide a more comprehensive assessment.

## Connection to existing approaches addressing failure modes
The paper identifies various failure modes across different PDE settings, which is valuable. However, there exist recent works that explicitly aim to address similar challenges, such as improving out-of-distribution generalization or long-term prediction stability. For instance, methods such as CNO demonstrate improved robustness under distribution shifts, and recent studies on rollout strategies explore how to mitigate error accumulation over long time horizons.
Incorporating these works—either through additional experiments or through a more detailed discussion—would further enhance the impact of the paper. In particular, going beyond identifying failure cases and providing insights into how such issues might be addressed would make the contribution more informative and useful to the community.

---

> ### Author Response · Authors · 2026-03-20
> **Response to reviewer uEgH**
>
> Thank you for your review. On the concern about whether the evidence sufficiently supports our claims, we believe the results are very strong. They are based on a large evaluation of 200 seeds per PDE with 1000 trained models in total, and the conclusions reflect consistent patterns across this full distribution rather than isolated runs. We will revise the manuscript to make this statistical grounding more explicit.
>
> We agree that extending the framework to models like DeepONet would be a valuable next step, and we will present this as a natural direction for future work while clarifying that the current claims are scoped to the studied setting and do not necessarily generalize across all neural operator architectures.
>
> We also agree on expanding the related work and adding spatial error visualizations, and will incorporate both in the revision. We will similarly expand the discussion to better connect the observed failure modes to existing approaches, and refine the wording throughout to more clearly reflect the paper's diagnostic rather than methodological scope.

---

> ### Author Response · Authors · 2026-04-11
> **Response to Reviewer uEgH (Revision Update)**
>
> Thank you again for your thoughtful and constructive feedback. We have made substantial revisions addressing your concerns, and the paper is significantly stronger as a result.
>
> Most significantly, we expanded the scope beyond FNO by incorporating two additional neural operator models (DeepONet-style and CNO) under the same stress-testing framework. This allows us to directly assess whether the observed failure modes are specific to FNO or shared across architectures, and to distinguish between architecture-dependent and more general patterns..
>
> In addition, we expanded the related work section to better reflect recent developments in neural operator research and strengthened the discussion connecting our findings to existing approaches that aim to address robustness and stability issues.
>
> Finally, we clarified the role of the degradation factor as a normalized robustness metric and its use alongside complementary diagnostics such as spectral and rollout analysis.
>
> We appreciate your suggestions, which directly shaped these revisions and improved both the clarity and impact of the paper.

---

### Review · Reviewer_oL7P · 2026-04-21

**Summary Of Contributions:**

# Summary
This paper proposes a stress-testing framework to evaluate the generalization ability of  neural PDE solvers under structured shifts. It is instantiated across three representative neural operator architectures on five PDE families. The experimental results shows that strong in-distribution accuracy does not reliably predict robustness under shift, emphasizing the importance of robustness under structured shift as the first-class evaluation target.

# Strengths
- The topic is interesting and important.
- The experimental setting is detailed.
# Weaknesses
- The paper studies generalization under distribution shift, which is closely related to the broader literature on domain generalization (DG) and domain adaptation (DA). However, the connection to these research areas is not sufficiently discussed. As a result, the conceptual positioning of the work remains somewhat narrow, and the paper misses an opportunity to clarify what is specific to neural operators and PDE settings versus what is shared with the broader DG/DA literature.
- Since the paper is primarily positioned as an evaluation/benchmarking study rather than a new method paper, the range of evaluated models appears rather limited. Only three baseline architectures are considered, which may not be sufficient to support broad conclusions about robustness in neural PDE solvers more generally.
- The analysis does not go sufficiently deep. The paper reports several interesting findings. For example, FNO achieving the best baseline accuracy yet showing poor robustness under certain Poisson shifts, and DeepONet underperforming in-distribution while being more robust under Black–Scholes payoff shift. But these are mostly presented as empirical observations rather than being explained in a mechanistic way.

**Audience:**

Yes

**Audience Explanation:**

The findings of this paper would be interesting to researchers in neural PDEs and transfer learning.

**Claims And Evidence:**

Yes

**Claims Explanation:**

The experiments support the claims.

**Requested Changes:**

To address the above weaknesses, I suggest that the authors:
- Discuss the literature on domain generalization and domain adaptation more explicitly, and clarify what aspects of the problem are specific to neural PDEs versus shared with these broader lines of work.
- Include additional strong baseline architectures, or otherwise better justify the current baseline selection and explain whether it is sufficiently representative of this area.
- Provide deeper analysis of the reported findings, for example by explaining the possible causes of these phenomena or conducting ablation studies to support such explanations.

---

> ### Author Response · Authors · 2026-04-21
> **Response to reviewer oL7P**
>
> Thank you for your reviews and the suggestions you've made.
>
> We agree that the connection to DG and DA should be made more explicit. The paper already frames the problem as structured generalization under shift, but we will clarify more directly what is shared with DG and DA and what is specific here, namely learned operators between function spaces together with shifts in coefficients, boundary or terminal conditions, discretization, and rollout horizon.
>
> On the model range, the current version studies three representative architectures under a common protocol, namely FNO, a DeepONet-style model, and CNO, across five PDE families, with 50 seeds per architecture and PDE pair and 750 trained models total. We agree that this is not exhaustive, and we do not claim universal coverage of all neural PDE solvers. The intended contribution is a comparative diagnostic framework that goes beyond single-model or single-equation evaluation and identifies both shared and architecture-dependent robustness patterns. We will make that scope clearer.
>
> We agree that the analysis can be deepened. The revised paper already makes several cross-setting findings more explicit, including the Poisson ranking reversal, the shared rollout instability in Navier Stokes, and the payoff-shift sensitivity in Black Scholes. We will strengthen the discussion of possible causes and connect these patterns more explicitly to architecture-specific inductive biases and PDE structure, while avoiding stronger mechanistic claims than the current experiments support.

---

> > ### Author Response · Authors · 2026-04-24
> > **Update to the requested changes by reviewer oL7P**
> >
> > Thank you again for your feedback. I’ve uploaded an updated draft that addresses the main points you raised. I expanded the related work section to connect the paper more clearly to domain generalization and domain adaptation, and to explain what’s shared with those settings versus what’s specific here. I also added a brief note on how this kind of benchmark could be useful for future AI4PDE foundation model directions. I clarified the reason the models given were selected by explaining that the three architectures represent 3 different inductive biases which give a representative sample from the many neural PDE solvers available. I strengthened the analysis of the results by connecting the main failure patterns more directly to the structure of the underlying PDEs and the biases of the architectures, and I added short theoretical propositions and proofs for two of the main stress mechanisms, namely rollout instability and resolution error. I appreciate your suggestions and believe the revised draft is much better as a result.

---

### Decision · Action_Editor_EidM · 2026-05-13

**Recommendation:** Accept as is

**Audience:**

Yes

**Audience Explanation:**

All reviewers agreed that the paper would be of interest some readers interested in neural PDEs, due to the additional and systematic empirical evidence on robustness which takes up on the previous literature.

**Claims And Evidence:**

Yes

**Claims Explanation:**

Overall, the reviewers agreed that the presented results are sound. During the review process both the framing of the claims and the breadth of the presented results where improved, thereby addressing reviewer concerns.

The remaining criticism of the paper was that its main contribution is the larger number and more systematic nature of empirical experiments, while not providing additional insights, deeper analysis of failure modes, or directions for mitigation. While the contribution may therefore be limited, the paper also does not claim to provide such deeper insights.